# SARS-CoV-2 Serum Viral Load and Prognostic Markers Proposal for COVID-19 Pneumonia in Low-Dose Radiation Therapy Treated Patients

**DOI:** 10.3390/jcm12030798

**Published:** 2023-01-19

**Authors:** Berta Piqué, Karla Peña, Francesc Riu, Johana C. Acosta, Laura Torres-Royo, Barbara Malave, Pablo Araguas, Rocío Benavides, Gabriel de Febrer, Jordi Camps, Jorge Joven, Meritxell Arenas, David Parada

**Affiliations:** 1Molecular Pathology Unit, Department of Pathology, Hospital Universitari Sant Joan de Reus, Institut d’Investigació Sanitàri Pere Virgili, Universitat Rovira i Virgili, 43204 Tarragona, Spain; 2Department of Radiation Oncology, Hospital Universitari Sant Joan de Reus, Institut d’Investigació Sanitàri Pere Virgili, Universitat Rovira i Virgili, 43204 Tarragona, Spain; 3Department of Geriatric and Palliative Care, Hospital Universitari Sant Joan de Reus, 43204 Tarragona, Spain; 4Unitat de Recerca Biomèdica, Institut d’Investigació Sanitària Pere Virgili, Universitat Rovira i Virgili, 43003 Tarragona, Spain

**Keywords:** COVID-19, pneumonia, low-dose radiation therapy, viral, load, RNA, prognostic, markers

## Abstract

Several studies have shown that the plasma RNA of SARS-CoV-2 seems to be associated with a worse prognosis of COVID-19. In the present study, we investigated plasma RNA in COVID-19 patients treated with low-dose radiotherapy to determine its prognostic value. Data were collected from the IPACOVID prospective clinical trial (NCT04380818). The study included 46 patients with COVID-19 pneumonia treated with a whole-lung dose of 0.5 Gy. Clinical follow-up, as well as laboratory variables, and SARS-CoV-2 serum viral load, were analyzed before LDRT, at 24 h, and one week after treatment. The mean age of the patients was 85 years, and none received any of the SARS-CoV-2 vaccine doses. The mortality ratio during the course of treatment was 33%. RT-qPCR showed amplification in 23 patients. Higher mortality rate was associated with detectable viremia. Additionally, C-reactive protein, lactate dehydrogenase, and aspartate aminotransferase were significant risk factors associated with COVID-19 mortality. Our present findings show that detectable SARS-CoV-2 plasma viremia 24 h before LDRT is significantly associated with increased mortality rates post-treatment, thus downsizing the treatment success.

## 1. Introduction

As of November 2022, Coronavirus disease 2019 (COVID-19), the highly transmissible viral illness caused by severe acute respiratory syndrome coronavirus 2 (SARS-CoV-2), has affected more than 600 million people and caused over 6 million deaths worldwide (https://COVID19.who.int/, accessed on 10 November 2022). Even though the efforts of the entire scientific community have led to unquestionable progress in unraveling the pathogenesis of SARS-CoV-2, and together with mass vaccination campaigns, the persisting spread of the virus has been limited, the emergence of potential new variants able to threaten the global public health again is an issue of concern.

SARS-CoV-2 is an enveloped, positive-sense, single-stranded RNA virus from the Coronaviridae family [1,2]. Although the mortality rate of SARS-CoV-2 is around 2.3%, while other members of the family, such as MERS-CoV, present an approximate mortality rate of 35% [3], it spreads fasters than its viral antecessors. The SARS-CoV-2 genome presents 14 open reading frames (ORFs) encoding 31 proteins, of which 16 are non-structural, 4 are structural proteins: spike (S), envelope (E), membrane (M), and nucleocapsid (N), and 11 are accessory proteins [4,5]. The clinical manifestation of COVID-19 is wide, ranging from mild illness to severe symptoms; however, the primary underlying cause of death of infected patients is respiratory failure [6]. Viral pneumonia is the most common indication for hospital admission and can lead to pulmonary dysfunction and acute respiratory distress syndrome (ARDS) [7]. COVID-19 severity is mainly attributed to a hyper-inflammatory response that leads to excessive secretion of pro-inflammatory cytokines [8,9].

Thus far, considerable progress in the therapeutic management of hospitalized patients has been made; however, there is still a significant gap in our knowledge to treat those patients that are not eligible for pharmacological intervention or mechanical ventilation, as described in the official guidelines [10]. In this scenario, some authors highlight low-dose radiotherapy (LDRT) as a novel therapeutic approach to treat COVID-19 pneumonia due to its anti-inflammatory properties [11,12,13,14]. The aim of the present study is to analyze how plasma viral load determination at the moment of irradiation, along with other clinical and laboratory factors, could help to predict the effectiveness of LDRT treatment in terms of survival in severe COVID-19 patients.

## 2. Materials and Methods

### 2.1. Participant Enrollment and Study Design

In this prospective and descriptive cohort study conducted at the Sant Joan University Hospital, Reus, Spain, we enrolled 46 hospitalized patients who tested positive for SARS-CoV-2 by RT-PCR from the IPACOVID clinical trial (NCT04380818) between June 2020 and February 2021. Inclusion criteria included patients with a positive COVID-19 diagnosis with moderate to severe pneumonia confirmed by chest X-ray requiring hospitalization with supplemental O_2_, who, due to comorbidities or general status, were not eligible for admission to the intensive care unit (ICU). Patients were treated with LDRT, specifically a single dose of 0.5 Gy to the whole thorax, during the acute phase of COVID-19 infection. The detailed protocol regarding the radiotherapy plan and sample collection has been previously published [11]. The present study was approved by the Institutional Review Board (IRB) of the Sant Joan University Hospital in Reus. Written informed consent was obtained from each participant in accordance with the recommendations established in the latest version of the Declaration of Helsinki, Fortaleza 2013.

The primary endpoint of the present study was to study baseline SARS-CoV-2 serum viral load, whether as a continuous or a categorical variable, and its gene profiling among patients in an LDRT-treated cohort. In this line, we wanted to analyze if pre-treatment serum viral load could be used as a predictor of infection mortality, either on its own or in combination with other relevant clinical and laboratory parameters that have been previously related to worse outcomes. The secondary aim was to investigate the direct association between SARS-CoV-2 serum viral load and the severity score CURB-65 (based on age, urea level, vital signs, and presence of confusion) and of the latter with the inflammatory blood marker interleukin-6 (IL-6). The third objective was to study if serum SARS-CoV-2 viremic individuals presented higher circulating total RNA serum concentration compared to aviremic patients. The fourth aim was to investigate whether differences were found in the evolution of C-reactive protein (CRP), aspartate aminotransferase (AST), and lactate dehydrogenase (LDH) concentrations throughout the 30-day follow-up of the trial between the serum SARS-CoV-2 positive group and the serum SARS-CoV-2 negative group. Finally, we wanted to examine the correlation between SARS-CoV-2 serum viral load and other COVID-19 severity risk factors.

### 2.2. Blood Samples

Peripheral blood samples of 5 mL were obtained from each hospitalized patient at four different time points: right before and 24 h, one week, and one month after LDRT. After collection, blood was left undisturbed for 20–30 min at room temperature to allow the clotting. Subsequently, tubes were centrifuged at 1000× *g* for 10 min to remove the clot, whereas the serum was transferred into 0.5 mL cryovials and immediately stored at −80° until use. Blood samples were used to determine serum SARS-CoV-2 viral load and biochemical markers.

### 2.3. RNA Serum Extraction

RNA was extracted from 500 µL of serum by using the High Pure Viral Nucleic Acid Large Volume Kit (Roche, Basel, Switzerland), according to the manufacturer’s instructions. In summary, serum samples were mixed with a working solution of binding buffer containing proteinase K and poly-A. Following 15 min of incubation at 70°, samples were transferred into a spin column inside a 50 mL tube and centrifuged for 5 min at 4000 rpm discarding the filtered liquid. Afterward, the spin column was placed in an elution tube and subjected to several washing steps. Finally, sample RNA was eluted in 50 mL of elution buffer, of which 10 µL were added to real-time RT-PCR Master Mix for amplification.

### 2.4. RT-PCR and SARS-CoV-2 Viral Load Quantification

SARS-CoV-2 serum RNA was amplified by RT-PCR using the TaqPath 1-Step Multiplex Master Mix commercial kit (Thermo Fisher Scientific, Waltham, MA, USA). This RT-PCR assay for SARS-CoV-2 includes forward and reverse primers and probes specific to three SARS-CoV2 genomic regions in the ORF1ab, Nucleocapside (N), and Spike (S) genes. MS2 phage ribonucleoprotein RNA was used as an internal process control for RNA extraction and amplification. RNA was quantified using a NanoDrop spectrophotometer (Thermo Fisher Scientific) by analyzing 2 µL of each sample. Samples were considered RT-qPCR SARS-CoV-2 positive if two of the three genes were amplified together with a Ct < 37 or the ORF or S genes were amplified alone with a Ct < 35. When the N gene was amplified alone, samples were considered negative for SARS-CoV-2 following the Interpretive Software determined by the manufacturer (Thermo Fisher Scientific).

SARS-CoV-2 RNAemia was quantified using a standard curve generated with known concentrations of viral load standards. The standard was the TaqPathTM COVID-19 RNA positive control provided by the TaqPath 1-Step Multiplex Master Mix commercial kit. 10-fold serial dilutions of the SARS-CoV-2 RNA positive control were used, setting the detection limit to approximately 4 copies/mL. Viral load expressed as log10 copies/mL was calculated by interpolating the Ct value of each sample into the ORF1ab gene standard curve, given that it was consistently amplified in all positive samples and was less likely to change among COVID-19 variants presenting spike protein mutations. Each RT-PCR assay was run with its specific standard curve, and each sample was analyzed in duplicate.

### 2.5. Statistical Analyses

Wilcoxon rank-sum test was used to compare continuous clinical variables, and Fisher’s exact test was used to assess differences between categorical variables. Actuarial survival analysis was calculated with Kaplan-Meier curves and a log-rank test. We performed Cox proportional hazards regression for COVID-19 mortality to calculate hazard ratios (HR) with 95% confidence intervals (CI) of different predictor variables. Significant variables from the univariant analysis were then included in a multivariant analysis. In order to do a broad screening of factors, the predictive efficacy for COVID-19 mortality in LDRT-treated patients was measured with receiver operating characteristic (ROC) curves and the area under the curve (AUC) for all the significant independent risk factors obtained in the univariant model. In order to study the combination of these individual risk factors with the aim of increasing the efficacy for predicting COVID-19 mortality, new variables combining two predictors were presented in the form of predicted probabilities obtained from a bivariant logistic regression model. These new combined predictors were analyzed with ROC analysis. Spearman’s Rho test was employed to analyze the correlation between two quantitative variables. Continuous variables are expressed as means and standard deviation, and categorical variables are expressed as frequencies and percentages. Statistical significance was set at *p* < 0.05. Statistical analyses, logistic regression, and graph representations were performed on SPSS Statistics version 25 (IBM SPSS Statistics, Armonk, NY, USA), GraphPad Prism 8.01 (GraphPad Software, San Diego, CA, USA), and NCSS Statistical Software 2022 version 12.0.10 (NCSS software, Kaysville, UT, USA).

## 3. Results

### 3.1. Clinical Findings

Between June 2020 and February 2021, 46 COVID-19-positive, 25 male and 21 female patients with a median age of 85 years, and all received dexamethasone treatment. None of the patients received any of the SARS-CoV-2 vaccine doses. Participants were treated with LDRT, specifically a single dose of 0.5 Gy to the whole thorax, during the acute phase of viral infection. The mortality ratio during the course of treatment was 33%. Table 1 summarizes the main clinical characteristics and laboratory findings categorized by survival at baseline.

### 3.2. RT-PCR and SARS-CoV-2 Viral Load Quantification Findings

Serum samples were collected at baseline and 1, 7, and 30 days after LDRT treatment was performed. Regarding gene distribution at baseline, serum RT-qPCR showed amplification for the ORF1ab, N, and/or S gene in 23 (50%) patients out of 46. 9 (39.13%) patients presented ORF, N and S gene amplification. One gene amplification was observed in 1 (4.35%) patient for the ORF gene, in 7 (30.43%) patients for the N gene, and in 2 (8.70%) patients for the S gene. ORF and S gene amplification was reported in 1 (4.35%) patient, ORF and N amplification was found in 3 (13.04%) patients, and none of the participants was positive for both N and S genes (Figure 1a). However, from these 23 samples that showed general amplification and following the standards of our laboratory, samples with the S and ORF1ab gene amplified alone with a Ct > 35 and the N gene amplified alone with no consideration of the Ct value were considered negative. Consequently, only 14 out of 46 participants (30.43%) exhibited SARS-CoV-2 serum RNA above the quantification limit (1 log10 copies/mL) at the time of basal blood collection and were considered viremic or SARS-CoV-2 RNA detectable participants. Among those individuals, the median serum viral load was 2.678 log10 RNA copies/mL (range 1.345–5.283 log10 RNA copies/mL; Figure 2a). Serum SARS-CoV2 viral load was reported as a continuous and categorical variable (detectable vs. undetectable), categorized as undetectable in all the samples below the quantification range.

### 3.3. LDRT and SARS-CoV-2 Viral Load Quantification Findings

Of the 46 participants studied, 31 (67.39%) patients recovered during the LDRT treatment, and 15 patients died (32.60%), of which 12 (26.09%) due to COVID-19 pneumonia and 3 (6.52%) due to other causes. When we considered the percentage of deaths, regardless of the cause, a higher mortality rate was associated with detectable viral load, as 57% of those patients with detectable serum viral load died compared to 22% in the undetectable viral load group (Fisher’s exact *p* = 0.0377; Figure 1c). Regarding COVID-19 mortality, serum viremia was also associated with increased mortality, as 57% of patients testing positive for SARS-CoV-2 serum RNA died compared to only 14% of those with undetectable viral load (Fisher’s exact *p* = 0.0085; Figure 1d). When analyzing viral load as a continuous variable, higher serum viral loads were significantly associated with mortality (Wilcoxon rank sum test *p* = 0.0068; Figure 1e), with a median viral load of 1.345 log10 RNA copies/mL (range 0–3.866 log10 RNA copies/mL) in the deceased group compared to a median viral load of 0 log10 RNA copies/mL (range 0–5.283 log10 RNA copies/mL) in the recovered group. Furthermore, patients were classified according to their CURB-65 score. Eleven (23.91%) patients presented a CURB-65 score of 2, 26 (56.52%) patients of 3, and 9 (19.56%) patients with a score of 4. The proportion of deceased patients with a CURB-65 score of 4 (77.8%) was significantly higher compared to patients with a score of 3 (26.9%) and a score of 2 (9.1%, Chi-square test *p* = 0.0032; Figure 1f). In this line, greater CURB-65 scores were correlated with SARS-CoV-2 plasma RNAemia (Figure 1g; Kruskal-Wallis *p* = 0.0006) and with higher levels of the inflammatory marker IL-6 (Figure 1h; Kruskal-Wallis *p* = 0.041). While the proportion of patients with SARS-CoV-2 plasma RNAemia was significantly associated with mortality and illness severity, no significant association was found between the length of hospitalization and the presence of SARS-CoV-2 plasma RNAemia among participants (Figure 1i). Moreover, in the recovered group, there is no significant association between the length of medical discharge and the presence of SARS-CoV-2 plasma RNAemia (Figure 1j). Considering circulating viral RNA detected in the 46 serum samples from the participants, no significant differences were found between serum SARS-CoV-2 positive (378.7 ± 33.21 ng/mL) and negative patients (385.9 ± 51.32 ng/mL) (Figure 1).

In this line, Kaplan-Meier survival analysis was used to estimate the survival time of patients in both the viremic and the aviremic groups, considering exclusively COVID-19 pneumonia as the cause of death. Kaplan-Meier analysis reported that those patients presenting SARS-CoV-2 serum positivity at admission survived less than negative SARS-CoV-2 serum patients. With a median follow-up of 15 days after hospital admission (range 0–57 days), the 30-day overall survival (OS) estimation was 54.4% for the entire studied cohort (Figure 2a). According to SARS-CoV-2 RNA serum presence, 30-day OS was 72.6% for negative serum SARS-CoV-2 patients and 30.5% for positive serum SARS-CoV-2 individuals (log-rank test *p* = 0.043; Figure 2b). Thus, it is confirmed that there is a significant association between COVID-19 serum RNA positivity and higher mortality rates.

In order to validate the usefulness of COVID-19 serum positivity one day before LDRT as a predictor of mortality, we wanted to analyze whether an association exists with some of the most relevant clinical and laboratory parameters that have been previously related to worse outcomes and disease severity. The univariate Cox proportional hazards regression analysis showed that viral load, C-reactive protein (CRP), lactate dehydrogenase (LDH), and aspartate aminotransferase (AST) were significant risk factors associated with COVID-19 mortality. Moreover, we set a p-value cut-off point of 0.1 in the Wald Test for identifying more potential candidates for further analysis, resulting in the addition of the age factor to the multivariant model as well. Subsequently, these parameters were incorporated into the multivariate Cox proportional-hazard model, which revealed that age (aHR = 1.463 [95% CI: 1.062–2.014]; Cox regression analysis *p* = 0.020) and viral load (aHR = 1.738 [95% CI: 1.009–2.992]; Cox regression analysis *p* = 0.046) were independent significant predictors of survival (Table 2).

Afterward, we performed a receiver operating characteristics (ROC) curve analysis with the significant variables from the univariate Cox analysis in order to perform a more extensive screening of factors in addition to viral load and age. When considering individual single parameters, AST showed the best results as per area under the curve (AUC: 0.755, *p* = 0.003), followed by LDH (AUC: 0.7467, *p* = 0.0018) and viral load (AUC: 0.746, *p* = 0.0015) (Figure 3a). Then, we combined all the parameters with viral load and also the combination of AST and LDH since they gave the best AUC values (Figure 3b). New variables were generated in the form of predicted probability from a binary logistic regression model.

Combined parameters were considered in order to improve the diagnostic efficacy of our predictors in the differentiation between the survivors and the deceased group during the first 30 days after LDRT administration. When combining AST and LDH with the other factors, AUC values were not as optimal as when these factors were combined with viral load. The viral load and AST combination showed the best diagnostic efficiency (AUC: 0.8589, *p* = 0), followed by viral load and LDH (AUC: 0.8468, *p* = 0; Table 3) in predicting COVID-19 mortality during a period of 30 days after LDRT treatment. Thus, a personalized analysis of viral load and AST parameters for each patient could help to predict the survival probability after LDRT treatment and be useful to distinguish which patients can benefit more from this treatment.

Then, we analyzed longitudinal measurements of CRP, LDH, and AST concentrations in both groups, the viremic and the aviremic participants. Although statistical significance was not found, we observed that after LDRT administration and during the following 30 days, there is a trend toward homogenization of CRP, LDH, and AST concentration values between patients with SARS-CoV-2 detectable serum viremia and those without (Figure 4).

Finally, we studied the correlation between the significant factors from the univariant Cox regression analysis and viral load (Figure 5). There was a significant moderate correlation between CURB-65 and viral load (r: 0.511, *p* < 0.001) (Figure 5d) and a low significant correlation between LDH and viral load (r: 0.317, *p* = 0.032) (Figure 5a). In the case of CRP/AST/Age and VL (Figure 5b,c,e), no statistical significance was achieved.

## 4. Discussion

The present study appears to be the first to analyze baseline SARS-CoV-2 plasma viral load in 46 LDRT-treated patients before irradiation and to investigate the usefulness of this parameter as a predictor of COVID-19 mortality during a 30-day follow-up period after the LDRT treatment. Our present findings show that detectable SARS-CoV-2 plasma viremia 24 h before LDRT is significantly associated with increased mortality rates post-treatment, thus downsizing the treatment success. Furthermore, we demonstrate that the patients with the larger scores in CURB-65 are the ones with the highest SARS-CoV-2 plasma viral load and that this pneumonia severity scale is significantly associated with inflammation, specifically with the cytokine IL-6. Furthermore, we report a significant positive correlation between SARS-CoV-2 viral load and LDH. Lastly, we suggest a potential novel mortality factor combining viral load and AST values that should be further investigated.

On March 11th of the year 2020, the WHO declared COVID-19 as a pandemic characterized by a broad clinical spectrum, being severe pneumonia and respiratory failure the main cause of death [7]. In this scenario, with the absence of specific antiviral drugs and sanitary systems collapsing over the globe, LDRT was resurrected as a potential treatment against COVID-19. The beneficial effects of LDRT treatment, based on its anti-inflammatory properties, date back to the last century. LDRT showed real curative benefits in the management of a variety of clinical conditions in which inflammation plays an important role, such as epicondylitis [15], osteoarthritis [16], necrotizing abscesses [17], and pneumonia. Several studies show that LDRT improves respiratory parameters and lung involvement and reduces serum inflammation markers [13,18,19,20]. A comparative cohort study reported a reduction in mortality among moderate COVID-19 patients treated with LDRT compared to the control group [21]. Furthermore, a systematic review, including nine clinical trials, supports that LDRT improves clinical parameters, radiological findings, and mortality rates, while at the same time, no side effects of radiation, such as acute toxicity, are found [22]. Regarding the biochemical mechanisms underlying LDRT, a recent study has broadened our knowledge, reporting an increase in serum PON1 activity while reducing inflammatory markers [12]. Although LDRT has shown promising results, considering the small sample and the lack of a randomized design in the majority of these studies, together with the knowledge gap in the understanding of the mechanistic behind LDRT, further research must be done.

In light of the results achieved thus far, LDRT could be an effective therapeutic strategy in cases of moderate and severe COVID-19, although there are some aspects regarding safety and dose that should be further investigated. For this reason, in our study, we propose a predictive method to maximize treatment success in terms of survivability based on the detection of SARS-CoV-2 viremia at baseline and thus be able to determine the most suitable patients for LDRT treatment. Several studies reported that SARS-CoV-2 serum viral load is associated with a higher risk of death, systemic inflammation, and disease progression [6,23,24]. The results of the present study confirm that the detection of serum viremia is related to higher mortality rates, and when quantified, there is a significant relationship between an increase in SARS-CoV-2 serum viral load and higher scores in the CURB-65 scale. Since the benefits of LDRT have not yet been demonstrated conclusively, together with the fact that not all hospitals have a radiotherapy department able to carry out LDRT in controlled safety conditions, we suggest this method to target LDRT treatment to the groups that can benefit most from its results. Furthermore, we found that patients with higher CURB-65 scores before treatment had larger baseline concentrations of IL-6, which has been highlighted to play a key role in the cytokine storm characterizing COVID-19 patients [25]. Both CURB-65 [26] and IL-6 [27] have been pointed out as reliable prognosis markers of COVID-19 patient outcomes, suggesting that SARS-CoV-2 baseline viral load could also be an effective severity predictor because of the relationship, either direct or indirect, we have demonstrated with both CURB-65 and IL-6.

We also analyzed other potential risk factors associated with COVID-19 mortality in LDRT-treated patients. In the present study, the univariate Cox regression model revealed other risk factors, apart from baseline serum viral load, including age, higher serum concentrations of CRP, LDH, and AST, and higher CURB-65 scores at baseline, which was consistent with findings of recent studies [28,29,30,31]. However, when multivariate Cox regression was performed, only viral load and age remained significant risk factors. In spite of this, we wanted to explore broadly possible prognosis predictive factors, and therefore we chose the ones from the univariate analysis. When we explored the predictive capacity of COVID-19 mortality of these individual parameters, AST, LDH, and viral load at baseline showed the best AUC values. Furthermore, we investigated whether the combination of different factors would give better AUC values in discriminating between patients who died after LDRT treatment over the 30-day follow-up of the study and those who did not, and indeed when combining viral load and AST factors, we maximized AUC values (AUC = 0.847). Liver damage causes elevated levels of both aspartate aminotransferase (AST) and alanine aminotransferase (ALT), which have been reported in COVID-19 patients [32]. Lei et al. reported that of these two enzymes, AST, in particular, is the one most closely associated with mortality [33]. Finally, CRP, LDH, and AST serum concentrations were monitored during the 30-day follow-up of the trial, and although no statistical differences were found, there is a tendency towards homogenization after LDRT is performed between SARS-CoV-2 serum detectable and the SARS-CoV-2 non-detectable group, we hypothesize it could be due to LDRT effect however, further research should be done.

Different studies have shown the presence of proinflammatory Th1 and Th2 cytokines in the serum of SARS patients compared to healthy controls, with significantly higher concentrations of TNF-ß, IL-6, IL-8, IL-10 and IL-12 in the stage early stage of SARS-CoV infection [34,35]. In the present study, we confirmed the role of the T cell response through the cytokine IL-6. Decreased serum cytokine levels have been reported to correlate with recovery from SARS-induced pneumonia. On the contrary, elevated levels of IL-4, IL-5, and IL-10 have been associated with fatal cases of SARS [36]. This cytokine response has been demonstrated with other viruses. Thus, in MERS-CoV, the increased secretion of IL-1, IL-6, IL-8, IL-12, and IFN, has been documented as a consequence of an antiviral and inflammatory response [36]. In addition, it has been shown that in, MERS-CoV the cytokines IL-8 and IL-12 are produced in greater amounts compared to the response to SARS-CoV [37]. It is interesting to highlight that increased plasma IL-6 concentration in SARS patients has been documented to be significantly increased in severe cases but not in convalescent or control subjects, suggesting a positive correlation between serum IL-6 level and disease severity [37]. Our findings confirm that plasma IL-6 levels are associated with severe COVID-19 disease and that IL-6 levels decrease in the convalescent period.

Our study has some limitations. First and most important, a control group could not be assembled. This control cohort would have consisted of patients receiving the standard of care (SoC) treatment without LDRT, enabling a comparative analysis of plasma COVID-19 viral load evolution, mortality ratio, length of hospitalization, viral RNA, and IL-6 concentration with the experimental cohort treated with SoC plus LDRT. Moreover, a longitudinal analysis of plasma SARS-CoV-2 viral load dynamics after LDRT could have provided more value to the results reported; however, our study was limited due to early discharges or COVID-19 negativization that did not facilitate further blood draws. In future studies, we would like to determine a cut-off point for the potential novel prognostic factor combining baseline AST concentration and plasma viral load that we have already pointed out. Furthermore, sample size was limited owing to difficulties encountered when matching different patient characteristics to form a homogenous group. Moreover, the limited number of patients in the hospital fitting in our inclusion criteria, along with the obstacles for every single one to be included in the trial, made it challenging to increase our sample size. On the other hand, the main strength of the present study lies in the fact that it is the first one to analyze baseline SARS-CoV-2 plasma viral load and to propose this parameter to be a good predictor for COVID-19 mortality in a LDRT-treated cohort. This would allow us to specifically treat patients that can benefit the most. In addition, we have demonstrated the association between plasma SARS-CoV-2 viral load and CURB-65 and LDH and, at the same time, the correlation between CURB-65 higher scores and larger IL-6 concentrations. Finally, in the present study, we highlight a novel COVID-19 mortality predictor combining viral load and AST concentrations that should be further investigated. Currently, it might appear that focusing efforts to further understand LDRT treatment is not required since COVID-19 progress is being restrained. However, we consider it important to broaden our knowledge on LDRT effects and underlying mechanisms due to new variables may emerge in world areas where massive vaccine campaigns are not being implemented, and also because we consider that LDRT treatment could open a new therapeutic window not only for COVID-19 treatment but also in several diseases where inflammatory response plays a role.

## Figures and Tables

**Figure 1 jcm-12-00798-f001:**
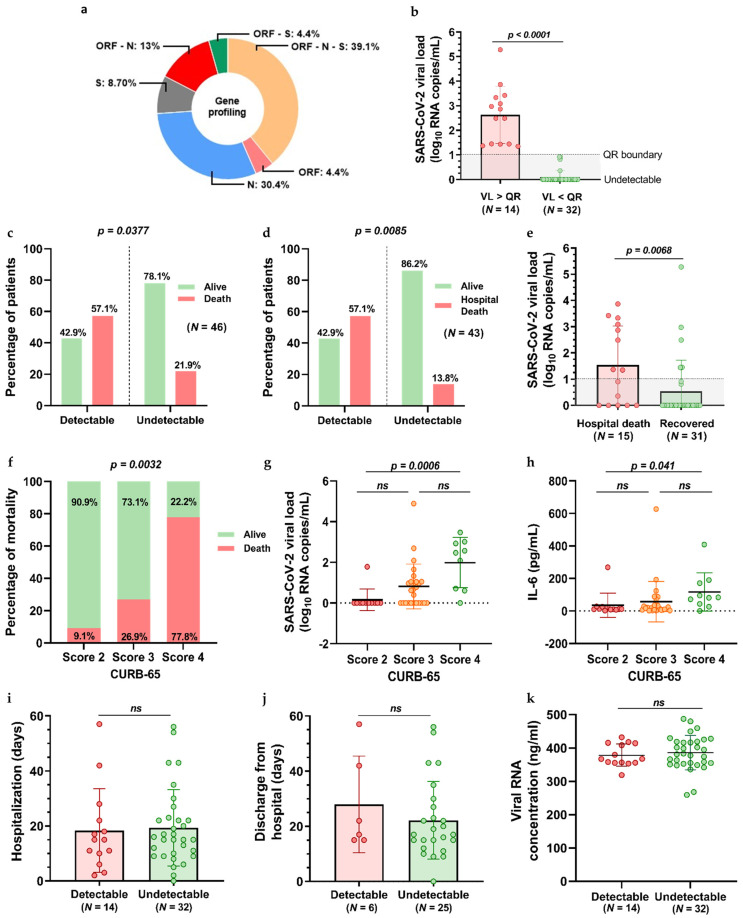
Gene profiling and SARS-CoV-2 RNA detection in serum at baseline before LDRT is administered. (**a**) Distribution of N, S, and ORF1ab gene amplification by qRT-PCR was detected in 23 patients, from which 14 were subsequently considered positive for SARS-CoV-2 following the guidelines of our laboratory. (**b**) SARS-CoV-2 viral load (VL) distribution; 4 patients presented viral loads below the range of detection and were considered undetectable throughout the trial. *p* values are from a two-tailed Wilcoxon rank sum test. (**c**,**d**) Percent of participants with detectable and undetectable serum SARS-CoV-2 viral load by mortality outcome, with (**d**) or without (**c**) consideration of COVID-19 pneumonia as the cause of death. *p* values are from a Fisher’s exact test. (**e**) Quantification of SARS-CoV-2 serum viral load at baseline in recovered and dead patients. (**f**) Distribution of patients’ mortality according to their CURB-65 scores. (**g**) Distribution of SARS-CoV-2 plasma viral load of patients according to their CURB-65 scores. (**h**) Interleukin-6 concentration corresponding to different CURB-65 scores. (**i**,**j**) Length of hospitalization (**f**) and medical discharge in detectable and undetectable SARS-CoV-2 serum patients. (**k**) Serum viral RNA concentration of participants. *p* values obtained from a two-tailed Wilcoxon rank sum test. ns stands for non-significant.

**Figure 2 jcm-12-00798-f002:**
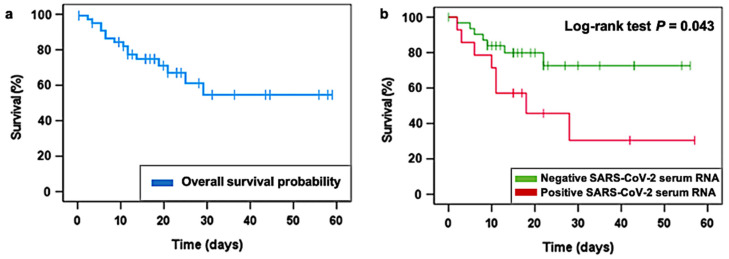
Overall survival (OS) estimates of COVID-19 patients treated with LDRT considering deaths due to COVID-19 pneumonia. (**a**) OS in the studied population (*N* = 47). (**b**) 30-day OS of patients testing positive for SARS-CoV-2 serum RNA (*N* = 14) and patients testing negative for SARS-CoV-2 serum RNA (*N* = 29). *p* value obtained from a log-rank test.

**Figure 3 jcm-12-00798-f003:**
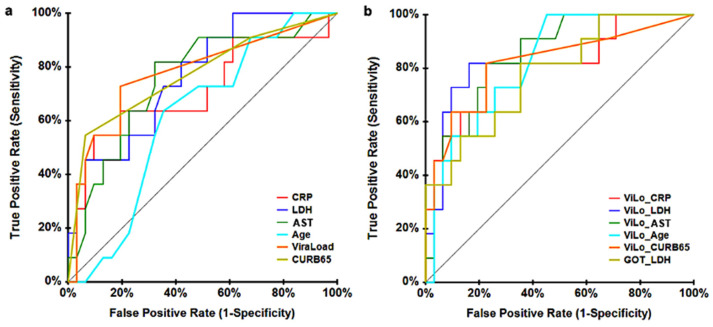
Receiver operating characteristics (ROC) curve analysis of different parameters in COVID-19 patients treated with LDRT to predict COVID-19 mortality. (**a**) ROC plot of single parameters (CRP, LDH, AST, Viral Load, CURB-65, Age). (**b**) ROC plot using a combination of parameters (Viral Load and C-reactive protein, Viral Load and LDH, Viral Load and AST, Viral Load and Age, Viral Load and CURB-65, AST and LDH).

**Figure 4 jcm-12-00798-f004:**
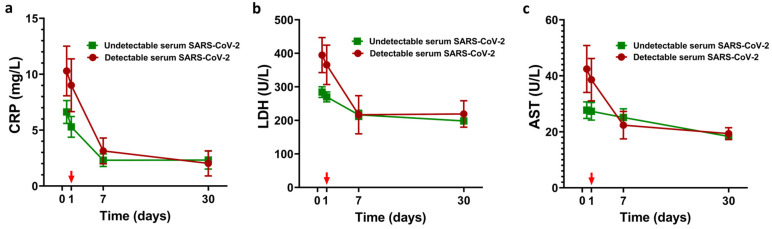
Longitudinal measurements of selected factors. Analysis of CRP (**a**), LDH (**b**), and AST (**c**) levels from baseline at day 0 to 30 days after LDRT. Treatment with LDRT was performed on day 1, indicated with a red arrow. Two-tailed Wilcoxon signed rank test did not show significant *p* values. SEM values are plotted.

**Figure 5 jcm-12-00798-f005:**
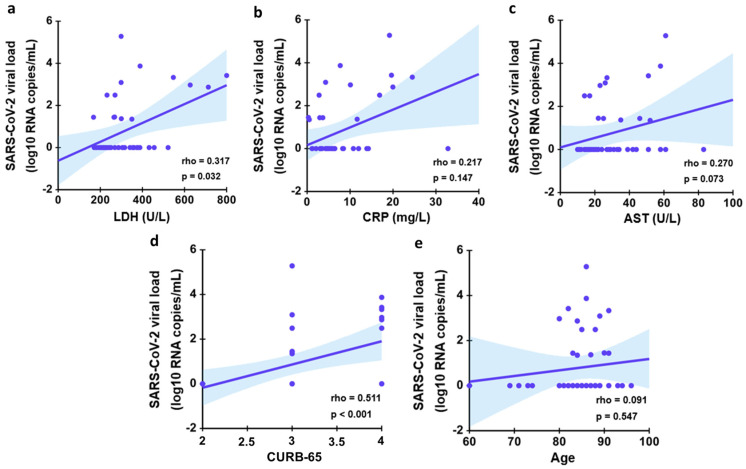
Correlation between viral load and other factors at baseline. Analysis of the association between LDH (**a**), CRP (**b**), AST (**c**), CURB-65 (**d**) and Age (**e**) with serum viral load. *p* values are obtained from Spearman correlation coefficient rho.

**Table 1 jcm-12-00798-t001:** Baseline demographic and clinical characteristics of COVID-19 patients.

Characteristic	Entire Cohort (*N* = 46)	Recovered (*N* = 31)	Deceased (*N* = 15)
Female	21 (45.65) ^b^	15 (48.39) ^b^	6 (40) ^b^
Male	25 (54.35) ^b^	16 (51.61) ^b^	9 (60) ^b^
Age	84.61 ± 6.611 ^a^	84.97 ± 5.351 ^a^	83.87 ± 8.847 ^a^
Comorbidities			
Neurological diseases	13 (28.26) ^b^	7 (22.58) ^b^	6 (40) ^b^
Cardiovascular diseases	37 (80.43)	25 (80.65)	12 (80)
Respiratory diseases	17 (36.96)	12 (38.71)	5 (33.33)
Other comorbidities	40 (86.96)	26 (83.87)	14 (93.33)
Pharmacological treatment			
Corticosteroids (dexamethasone)	46 (100) ^b^	31 (100)	15 (100)
Functional Status (Barthel Index)			
Independent	8 (17.39) ^b^	7 (22.58) ^b^	1 (6.67) ^b^
Minimally dependent	19 (41.30)	14 (45.16)	5 (33.33)
Partially dependent	9 (19.57)	4 (12.90)	5 (33.33)
Very dependent	7 (15.22)	5 (16.13)	2 (13.33)
Total dependent	3 (6.52)	1 (3.23)	2 (13.33)
Geriatric Depression Scale (GDS)			
No cognitive decline	23 (50) ^b^	19 (61.29) ^b^	4 (26.67)
Very mild cognitive decline	8 (17.39)	4 (12.90)	4 (26.67)
Mild cognitive decline	9 (19.57)	5 (16.13)	4 (26.67)
Moderate cognitive decline	1 (2.17)	1 (3.23)	0 (0)
Moderately severe cognitive decline	2 (4.35)	2 (6.45)	0 (0)
Severe cognitive decline	3 (6.52)	0 (0)	3 (20)
Very severe cognitive decline	0 (0)	0 (0)	0 (0)
Basal SpO_2_	93.43 ± 2.713 ^a^	94.03 ± 2.627 ^a^	92.20 ± 2.541 ^a^
Basal SaFi	283 ± 94.720 ^a^	305 ± 80.870 ^a^	237.40 ± 107.400 ^a^
Mild	37 (80.43) ^b^	28 (90.32) ^b^	9 (60) ^b^
Moderate	4 (8.70)	2 (6.45)	2 (13.33)
Severe	5 (10.87)	1 (3.23)	4 (26.67)
CURB-65 Score			
Score 1	-	-	-
Score 2	12 (26.09) ^b^	10 (32.26) ^b^	2 (13.33)
Score 3	25 (54.35)	19 (61.29)	6 (40)
Score 4	9 (19.56)	2 (6.45)	7 (46.67)
Score 5	-	-	-
First radiological findings			
CT lung involvement <5%	-	-	-
CT lung involvement 5−25%	1 (2.17) ^b^	−	1 (6.67)
CT lung involvement 26−49%	7 (15.22)	6 (19.35)	1 (6.67)
CT lung involvement 50−75%	22 (47.83)	17 (54.84)	5 (33.33)
CT lung involvement >75%	16 (34.78)	8 (25.81)	8 (53.33)

SaFi: ratio of oxygen saturation (SpO2) to fractional inspired oxygen (FiO2); CURB-65: clinical criteria validated to guide the treatment of community-acquired pneumonia based on confusion, BUN, respiratory rate, systolic blood pressure, and age; CT: computed tomography. ^a^ Data shown as means ± standard deviations. ^b^ Data shown as the number of patients and percentages in parenthesis.

**Table 2 jcm-12-00798-t002:** Bivariate and multivariate Cox proportional hazards regression model for COVID-19 mortality.

Risk Factors	Univariate Cox Regression Analysis		Multivariate Cox Regression Analysis	
	HR (95% CI)	*p* Value	HR (95% CI)	*p* Value
Age	1.125 (0.990–1.279)	0.071	1.463 (1.062–2.014)	0.020
Sex (Female)	0.897 (0.325–2.481)	0.835	-	-
Viral Load	1.596 (1.166–2.184)	0.004	1.738 (1.009–2.992)	0.046
RNA serum concentration	1.005 (0.991–1.019)	0.466	-	-
Neurological disease	1.542 (0.525–4.525)	0.431	-	-
Cardiovascular disease	1.355 (0.378–4.865)	0.641	-	-
Respiratory disease	1.111 (0.394–3.130)	0.843	-	-
Other comorbidities	2.578 (0.338–19.648)	0.361	-	-
Days with symptoms	1.075 (0.782–1.476)	0.657	-	-
IL-6	1.000 (0.995–1.005)	0.927	-	-
Ferritin	1.000 (1.000–1.000)	0.143	-	-
C-reactive protein	1.075 (1.017–1.138)	0.011	0.979 (0.841–1.141)	0.789
LDH	1.004 (1.001–1.007)	0.005	1.007 (1.000–1.014)	0.064
Hemoglobin	1.244 (0.953–1.625)	0.108	-	-
CD4 cells	0.998 (0.994–1.003)	0.486	-	-
CD8 cells	0.999 (0.994–1.004)	0.708	-	-
CD4/CD8 ratio	0.961 (0.774–1.192)	0.715	-	-
D-dimer	1.000 (1.000–1.000)	0.278	-	-
CURB-65	3.153 (1.348–7.373)	0.008	1.795 (0.398–8.081)	0.446
Glucose	1.002 (0.996–1.008)	0.485	-	-
Respiratory Frequency	1.000 (0.981–1.019)	0.986	-	-
AST	1.019 (1.005–1.034)	0.009	0.996 (0.974–1.018)	0.706
ALT	1.010 (0.992–1.029)	0.255	-	-
Leukocytes	1.000 (1.000–1.000)	0.951	-	-
Lymphocytes	0.999 (0.998–1.001)	0.348	-	-
Platelets	1.000 (1.000–1.000)	0.165	-	-

IL-6: interleukin-6; LDH: lactate dehydrogenase; CURB-65: clinical criteria validated to guide the treatment of community-acquired pneumonia based on confusion, BUN, respiratory rate, systolic blood pressure, and age; AST: aspartate aminotransferase; ALT: alanine aminotransferase.

**Table 3 jcm-12-00798-t003:** Area under the curve for various factors in COVID-19 patients.

Variable	AUC	*p* Value	95% Confidence Interval	
			Lower Bound	Upper Bound
Viral Load	0.7460	0.0015	0.5352	0.8693
CURB-65	0.7377	0.0019	0.5309	0.8616
LDH	0.7467	0.0018	0.5293	0.8720
AST	0.7552	0.0003	0.5670	0.8685
CRP	0.7313	0.0032	0.5184	0.8588
Age	0.6698	0.0201	0.4747	0.8022
Viral Load + CRP	0.7957	0.0001	0.5774	0.9078
Viral Load + LDH	0.8468	0.0000	0.6313	0.9409
Viral Load + AST	0.8589	0.0000	0.6894	0.9392
Viral Load + Age	0.8320	0.0000	0.6590	0.9214
Viral Load + CURB-65	0.8196	0.0001	0.5790	0.9297
AST + LDH	0.7768	0.0001	0.5846	0.8864

CURB-65: clinical criteria validated to guide the treatment of community-acquired pneumonia based on confusion; LDH: lactate dehydrogenase; AST: aspartate aminotransferase; CRP: C-reactive protein. New variables were generated by combining two factors in form of predicted probability from a binary logistic regression model.

## Data Availability

Data is contained within the article.

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
