# Peer review of "SARS-CoV-2 Serum Viral Load and Prognostic Markers Proposal for COVID-19 Pneumonia in Low-Dose Radiation Therapy Treated Patients"

_jcm, 2023, doi:10.3390/jcm12030798_

Round 1

Reviewer 1 Report

The manuscript needs more edits on the main-results detail. The images need to improve from resoulation. The conclusion should be modified based on the study results. In discussion compare the results with other coronaviruses such as SARS-CoV and MERS-CoV.

Author Response

Response to Reviewer 1:

Dear reviewer, thank you for your valuable comment and time. We have collected your comments in order to improve our work and so that it can be assessed for possible publication. Here are the changes made:

1) Point 1: The manuscript needs more edits on the main-results detail.

1.1) Response to point 1: Following your indication, we have made changes in the results section, applying subheadings in reference to the study findings, highlighting the results, for example line 157: clinical findings. We have also reviewed and homogenized the results, to emphasize the importance of the findings.

2) Point 2: The images need to improve from resolution.

2.1) Response to point 2: Taking into account your indication, we have improved the quality of all images, which are in TIFF format high resolution.

3) Point 3: The conclusion should be modified based on the study results

3.1) Response to point 3: Taking your point into account and since the conclusion is not a mandatory point, we have decided to eliminate the conclusion.

4) Point 4: In discussion compare the results with other coronaviruses such as SARS-CoV and MERS-CoV.

4.1) Response to point 4: Thank you for your comment and following your indication we have added the following information in the discussion:

Different studies have shown the presence of proinflammatory Th1 and Th2 cyto-kines in the serum of SARS patients compared to healthy controls, with significantly higher concentrations of TNF-ß, IL-6, IL-8, IL-10 and IL-12 in the stage early stage of SARS-CoV infection [41, 42]. In the present study we were able to confirm the role of the T cell response, through the cytokine IL-6. Decreased serum cytokine levels have been reported to correlate with recovery from SARS-induced pneumonia. On the contrary, elevated levels of IL-4, IL-5, and IL-10 have been associated with fatal cases of SARS [43]. This cytokine response has been demonstrated with other viruses. Thus, in MERS-CoV, the increased secretion of IL-1, IL-6, IL-8, IL-12, and IFN, has been documented as a consequence of an antiviral and inflammatory response [43]. In addition, it has been shown that in MERS-CoV the cytokines IL-8 and IL-12 are produced in greater amounts compared to the response to SARS-CoV [44]. It is interesting to highlight that Increased plasma IL-6 concentration of SARS patients has been documented to be significantly in-creased in severe cases, but not in convalescent or control subjects, suggesting a positive correlation between serum IL-6 level and the disease severity [44]. Our findings confirm that plasma IL-6 levels are associated with severe COVID-19 disease, and that IL-6 levels decrease in the convalescent period.

New references:

  1. Dosch, S.F.; Mahajan, S.D.; Collins, A.R. SARS Coronavirus Spike Protein-Induced Innate Immune Response Occurs via Activation of the NF-kappaB Pathway in Human Monocyte Macrophages in Vitro. Virus Res. 2009, 142, 19–27.
  2. Wu, H.; Yan, H.; Ma, S.; Wang, L.; Zhang, M.; Tang, X.; Temperton, N.J.; Weiss, R.A.; Brenchley, J.M.; et al. T Cell Responses to Whole SARS Coronavirus in Humans. J. Immunol. 2008, 181, 5490–5500.
  3. Sinderewicz, E.; Czelejewska, W.; Jezierska-Wozniak, K.; Staszkiewicz-Chodor, J.; Maksymowicz, W. Immune Response to COVID-19: Can We Benefit from the SARS-CoV and MERS-CoV Pandemic Experience? Pathogens 2020, 9, 739.
  4. Zhang, Y.; Li, J.; Zhan, Y.; Wu, L.; Yu, X.; Zhang, W.; Ye, L.; Xu, S.; Sun, R.; Wang, Y.; et al. Analysis of Serum Cytokines in Patients with Severe Acute Respiratory Syndrome. Infect. Immun. 2004, 72, 4410–4415.

Reviewer 2 Report

Thankyou for asking me to review this interesting paper on the role of serum SARS-COV-2 viral load in predicting mortality in a small cohort of unvaccinated, patients, over the age of 80 years old, not suitable for intensive care treated with LDRT. 

The introduction is concise and well presented. The methods are thorough and there is extensive detail on the experimental design, which helps the reader follow the narrative. The results are well presented and the statistical analysis is sound. The discussion is extensive and identifies some of the paper's limitations. There is a lot of discussion around the potential benefits of LDRT which is not a well established treatment for COVID-19 and is not in any International guidelines. This is mentioned in the limitations. There is no discussion of the small study and the very narrow inclusion criteria, which should be included in the limitations section. 

Furthermore the authors should comment on the adjustments they have made for multiple comparisons on such a small cohort. 

Overall the paper is well written and thought out but the narrow inclusion criteria (unvaccinated, over 80s, not suitable for ICU, treated with LDRT) reduce the generalisability and therefore relevance of this paper outside specific centres keen on LDRT for experimental purposes. If a control group were available this would add significant weight to the paper's usefulness. 

Minor points 

Exitus is Figure 1 should be changed to death 

LD-T should be LDRT line 370

If the conclusion section (section 5) is not to be used it should be deleted

Author Response

Response to Reviewer 2:

Dear reviewer, thank you for your valuable comment and time. We have collected your comments in order to improve our work and so that it can be assessed for possible publication. Here are the changes made:

1) Point 1: There is a lot of discussion around the potential benefits of LDRT which is not a well-established treatment for COVID-19 and is not in any International guidelines. This is mentioned in the limitations. There is no discussion of the small study and the very narrow inclusion criteria, which should be included in the limitations section.

1.1) Response to point 1: We deeply appreciate your comments about our manuscript.

Regarding the small size of our sample, we have added a paragraph explaining the limitations we encountered and that hindered the arrangement of a bigger sample group in the Discussion, concretely in the limitation paragraph:

Line 444-448: Furthermore, sample size was limited owing to difficulties encountered when matching different patient characteristics to form a homogenous group. Also, the limited number of patients in the hospital fitting in our inclusion criteria, along with the obstacles for every single one to be included in the trial, made it really challenging to increase our sample size.

Additionally, during the COVID-19 pandemic, our center was one of the hospitals selected as a reference center for the treatment of patients with COVID-19, and as you point out, our study focused mainly on patients over 80 years of age and without ICU admission criteria. , to which low-dose radiotherapy treatment could be applied. This treatment was agreed upon with other centers in the region and was considered a therapeutic option in this restricted group of patients.

As for the control group suggested by you, obtaining it was excluded because it could not be carried out to meet the diagnostic criteria and the use of LDRT. Regarding the usefulness of viral load, our previous work (Peña, K.B.; Riu, F.; Gumà, J.; Guilarte, C.; Pique, B.; Hernandez, A.; Àvila, A.; Parra, S.; Castro, A.; Rovira, C.; Cueto, P.; Vallverdu, I.; Parada, D. Study of the Plasma and Buffy Coat in Patients with SARS-CoV-2 Infection—A Preliminary Report. Pathogens 2021, 10, 805.) demonstrated its usefulness by comparing COVID-19 positive patients, COVID-19 negative patients, and a control group, demonstrating that viral load can be a prognostic and predictive biomarker in patients with COVID-19. 19

Finally, we have added the following information in the discussion:

Different studies have shown the presence of proinflammatory Th1 and Th2 cyto-kines in the serum of SARS patients compared to healthy controls, with significantly higher concentrations of TNF-ß, IL-6, IL-8, IL-10 and IL-12 in the stage early stage of SARS-CoV infection [41, 42]. In the present study we were able to confirm the role of the T cell response, through the cytokine IL-6. Decreased serum cytokine levels have been reported to correlate with recovery from SARS-induced pneumonia. On the contrary, elevated levels of IL-4, IL-5, and IL-10 have been associated with fatal cases of SARS [43]. This cytokine response has been demonstrated with other viruses. Thus, in MERS-CoV, the increased secretion of IL-1, IL-6, IL-8, IL-12, and IFN, has been documented as a consequence of an antiviral and inflammatory response [43]. In addition, it has been shown that in MERS-CoV the cytokines IL-8 and IL-12 are produced in greater amounts compared to the response to SARS-CoV [44]. It is interesting to highlight that Increased plasma IL-6 concentration of SARS patients has been documented to be significantly in-creased in severe cases, but not in convalescent or control subjects, suggesting a positive correlation between serum IL-6 level and the disease severity [44]. Our findings confirm that plasma IL-6 levels are associated with severe COVID-19 disease, and that IL-6 levels decrease in the convalescent period.

New references:

  1. Dosch, S.F.; Mahajan, S.D.; Collins, A.R. SARS Coronavirus Spike Protein-Induced Innate Immune Response Occurs via Activation of the NF-kappaB Pathway in Human Monocyte Macrophages in Vitro. Virus Res. 2009, 142, 19–27.
  2. Wu, H.; Yan, H.; Ma, S.; Wang, L.; Zhang, M.; Tang, X.; Temperton, N.J.; Weiss, R.A.; Brenchley, J.M.; et al. T Cell Responses to Whole SARS Coronavirus in Humans. J. Immunol. 2008, 181, 5490–5500.
  3. Sinderewicz, E.; Czelejewska, W.; Jezierska-Wozniak, K.; Staszkiewicz-Chodor, J.; Maksymowicz, W. Immune Response to COVID-19: Can We Benefit from the SARS-CoV and MERS-CoV Pandemic Experience? Pathogens 2020, 9, 739.
  4. Zhang, Y.; Li, J.; Zhan, Y.; Wu, L.; Yu, X.; Zhang, W.; Ye, L.; Xu, S.; Sun, R.; Wang, Y.; et al. Analysis of Serum Cytokines in Patients with Severe Acute Respiratory Syndrome. Infect. Immun. 2004, 72, 4410–4415.

Minor points:  

  • Exitus in Figure 1 has been changed to death
  • ine 364: LD-RT has been changed to LDRT.
  • If the conclusion section (section 5) is not to be used it should be deleted. Taking your point into account and since the conclusion is not a mandatory point, we have decided to eliminate the conclusion.